# Design and Simulation of a System-in-Package Chip for Combined Navigation

**DOI:** 10.3390/mi15020167

**Published:** 2024-01-23

**Authors:** Yang Yang, Guangyi Shi, Yufeng Jin

**Affiliations:** 1School of Electronic and Computer Engineering, Shenzhen Graduate School of Peking University, Shenzhen 518055, China; yang1994@pku.edu.cn; 2School of Software and Microelectronics, Peking University, Beijing 102600, China; shiguangyi@ss.pku.edu.cn

**Keywords:** system-in-package, combined navigation, signal integrity, power integrity, electrical-thermal-mechanical analysis

## Abstract

This paper proposes a system-in-package combination navigation chip. We used wire bonding, chip stacking, surface mount, and other processes to integrate satellite navigation chips, inertial navigation chips, microprocessor chips, and separation devices. Finally, we realized the hardware requirements for combined navigation in a 20 mm × 20 mm chip. Further, we performed a multi-physics simulation analysis of the package design. For antenna signals, the insertion loss was greater than −1 dB@1 GHz and the return loss was less than −10 dB@1 GHz. The amplitude of these noises of the signal between the MCU and the IMU was approximately 20%, and the maximum value of the coupling coefficient between signal lines on the top surface was 13.4174%. The ninth mode of the power plane yielded a maximum voltage of 55 mV, and all power delivery networks had a DC voltage drop of less than 2%. The highest temperature in the microsystem was approximately 42 °C. These results show that our design performed well in terms of signal, power, and thermal performance.

## 1. Introduction

As the cost of advanced process nodes becomes more and more expensive, the economic benefits of Moore’s Law have become invalid. Compared with the design complexity and low yield caused by integrating all functions on a single big silicon chip, system-in-package (SiP) provides a more flexible, efficient, and low-cost development direction that integrates multiple components, such as a CPU, digital logic, analog/mixed signals, memory, sensors, and passive and discrete components, within a single package and a single system. SiP has attracted much research in recent years, which promotes its application in the fields of high-performance computing (HPC), multi-sensor fusion, radio frequency, power electronics, and other fields [1,2,3,4,5]. With the continuous development of integrated circuit technology, electronic products are increasingly developing in the directions of miniaturization, intelligence, high performance, and high reliability. In this process, SiP plays an important role and is widely used [6,7,8].

System-in-package often faces multi-physics and multi-scale problems, which are key issues in their design and simulation [9,10,11,12]. When finite element analysis is performed on components with large size differences, such as chips, packages, and circuit boards, a large number of meshes are generated, which cause difficulty and usually require simplified and reduced-order processing; this problem, however, was not the focus of this study. In order to ensure that microsystems can operate well and be highly reliable, researchers have chronically paid attention to and analyzed their electrical, thermal, mechanical and other properties, as well as the coupling characteristics existing in them [13,14,15,16]. To achieve system-level performance, signal integrity and power integrity are inescapable issues and challenges [17,18,19]. The signal path between chips needs to be high-speed and low-noise, while the power plane needs to provide stable voltage and sufficient current. In addition, microsystems need good heat dissipation to avoid performance degradation and failures caused by high temperatures [20,21,22,23]. Temperature causes thermal stress, and temperature mismatch is the main cause of the warping of microsystem structures [24]. Warpage and stress are the main causes of microsystem failure which affects reliability and yield, so it has become a research hotspot and attracted countless scholars [25,26,27,28,29,30]. However, most of the above studies discuss only the single-physics problem, and fail to obtain the results of multi-physics coupling, which are obviously deviated from the actual situation.

In order to further study the multi-physical coupling characteristics of system-in-package, a SiP chip for integrated navigation was designed in this study. Combined navigation systems include satellite navigation chips, inertial navigation chips, microprocessor chips, flash chips, and discrete devices [31,32]. This system-in-package allows a combined navigation system to be packaged into a single chip, resulting in reduced system size, lower power consumption, and ease of application; we named it a system-in-package combined navigation chip microsystem. This paper is organized as follows. In Section 2, the design of the microsystem is introduced, including signal transmission, packaging process, and substrate layout. Section 3 focuses on the simulation analysis of the designed microsystem. The performance of the microsystem was evaluated in terms of signal integrity, power integrity, and electrical-thermal-mechanical analysis. In Section 4, we fully analyze the coupling effects between the multi-physics. Finally, we summarize relevant data and propose a more comprehensive system-in-package design and simulation method.

## 2. Design of the Combined Navigation Chip

In the scheme design, we wanted to integrate the core functionality of combined navigation in one package. Using SiP technology to integrate multiple silicon chips and their peripheral circuits can reduce the size of the system. The combined navigation mentioned here refers to the technology applied to assisted driving and autonomous driving, which achieves high-precision and reliable navigation through the fusion of satellite navigation and inertial navigation. The main data transmission flow diagram of the microsystem is shown below in Figure 1: two satellite navigation chips receive satellite signals from external antennas and transmit data to the microprocessor chip. The inertial navigation chip collects acceleration and angular acceleration data and also transmits it to the microprocessor chip, the microprocessor chip receives differential data from the outside to realize a satellite RTK solution and runs the combined navigation coupling algorithm, the final navigation result is output to other devices, and the flash chip is used to store data and programs.

The rectangular box in Figure 1 (above) shows the components we needed to integrate into the microsystem, and also includes some peripheral circuits that are not shown in the figure. The next step was to determine the package design. The capacitors, resistors, inductors, and crystal oscillators required for peripheral circuits were mounted on the surface of the substrate using surface mount technology. Microprocessor chip bare die, flash chip bare die, and satellite navigation chip bare die are all suitable for wire bonding package designs. The flash required for satellite navigation chips increased integration through chip stacking. The overall packaging process diagram is shown below in Figure 2.

After determining the microsystem integration components and packaging technical solutions, we preliminarily designed the layout of the microsystem according to the size and number of components, as well as the corresponding connection relationship. First, the two satellite navigation chips needed to share a set of crystal oscillators to ensure that their clocks are synchronized. We also needed to design an antenna signal path with 50 Ω impedance, so the two satellite navigation naked dies were placed side by side in the upper part of the substrate, and the required crystal oscillators were placed in the middle of the two satellite navigation chips. The microprocessor chip was relatively large in size and had the largest number of pins, so it was placed in the lower part of the middle of the substrate, and the smaller flash size connected to the microprocessor chip was placed on its left. The inertial navigation chip connection line was simple and was placed in the lower right corner of the substrate to maintain an appropriate distance from the microprocessor chip. The crystal oscillator required by the microprocessor chip was placed in its corresponding pin position, which was in the upper right corner of the microprocessor chip. Other required discrete components, such as capacitors, resistors, and inductors, were arranged around these main components. All devices were integrated on the surface of the substrate, which adopted a square design and was set to 20 mm × 20 mm in size; the corresponding layout diagram is shown below in Figure 3.

The parameters of each material layer of the four-layer organic substrate recommended by the packaging manufacturer follow in Table 1.

## 3. Multi-Physics Simulation Analysis of the System-in-Package

After completing the package design, we imported the design file into the simulation software, which allowed us to perform electrical and thermal simulations of the designed package structure. Detailed simulation analyses are described below, and relevant simulation data are provided in the data file.

### 3.1. Signal Integrity

In this section, we analyze the signal integrity of the package design from multiple perspectives. In the entire design, the MCU section occupies the largest number of signals. Its maximum operating frequency is 1 GHz, and we focused on analyzing the insertion loss of its signal path within 2 GHz. As shown below in Figure 4a, we simulated that the insertion loss of all signal lines of the MCU was greater than −2 dB within 2 GHz. In Figure 4b, the return loss of all signal lines of the MCU was less than −20 dB within 500 MHz. These data mean that the MCU’s signal quality was guaranteed such that its signal speed was less than 10 MHz.

In addition, regarding the satellite navigation chip, there are two special RF signal inputs. We analyzed its S-parameters within 30 GHz. It was observed that above 5 GHz its return loss significantly deteriorated. Fortunately, however, the input signal operated at 1–2 GHz, and the S-parameters in this part performed well, as shown below in Figure 5. The insertion loss was greater than −1 dB@1 GHz and the return loss was less than −10 dB@1 GHz. It can be seen that the quality of the RF signal was significantly better than that of the MCU, because we added a ring of ground holes around the RF signal line to provide a close reflow loop, and also isolate interference, as shown in Figure 5. At the same time, the RF signal line was also designed to be as short as possible to obtain good S parameters.

We further analyzed the time domain characteristics of the signal between the MCU and the IMU, and saw that the quality of the signal transmission was good. At the same time, we observed signal overshoot, as well as near-end and far-end signal noise. However, the amplitude of these noises was approximately 20% and did not produce logic errors, as shown below in Figure 6.

Figure 7, below, shows the coupling coefficient between the signal lines on the top surface, with a maximum value of 13.4174%. In the middle part of the layout, large coupling was caused by the small spacing of the signal lines. On the whole, such coupling is acceptable.

We conducted frequency domain or time domain analysis on multiple sets of data lines in the microsystem and they all met the working requirements, so it can be seen that the design meets signal integrity. The power supply performance of the microsystem is analyzed below.

### 3.2. Power Integrity

The good signal characteristics of SiP are shown above, and we further discuss its power integrity here. First, we analyzed the resonant characteristics of its power supply plane. Simulation results are shown below in Table 2, and we obtained nine resonant modes. The first three and last planar resonance diagrams are shown in Figure 8. The last five resonant modes all corresponded to the 3.3 V plane. The 9th mode yielded a maximum voltage value of 55 mV.

Further, we analyzed the impedance characteristics of a 1.8 V power supply at the MCU side. As can be seen below in Figure 9, before 500 MHz, its impedance was less than 1 Ω. There was a resonant front at high frequencies that could be further optimized using decoupling capacitors.

As shown above, the resonant noise of the power delivery network was small, and impedance characteristics were satisfied in the operating frequency. We also simulated the DC voltage drop characteristics of the power supply network, as shown in Table 3. All power delivery networks had a DC voltage drop of less than 2%, which met the requirements of the package design. In summary, the design of this microsystem is in line with power integrity requirements. In the following section, we discuss the electro-thermal-mechanical coupling analysis of the microsystem.

### 3.3. Electrical-Thermal-Mechanical Analysis

For system-in-package, the influence of temperature characteristics is important, so we performed an electro-thermal co-simulation, taking into account the effects of thermal stress, and at the same time were able to obtain results for three physics at once. We set up an analysis scenario, shown below in Figure 10, in which the SiP chip was mounted on a PCB test board with a heat dissipation structure at the bottom. The material and size parameters of the main components are shown in Table 4.

According to the actual chip power, the temperature distribution of this microsystem is shown below in Figure 11, and the highest temperature was in the MCU area because it has the highest power level. Due to mismatch in the coefficient of thermal expansion, the temperature caused warpage and thermal stress of the substrate. Therefore, we observed the largest displacement of pins (about 74 microns) in the four corners, as shown in the figure below. This was also the stress concentration area, which is prone to failure and was the focus area of design optimization, as shown below in Figure 12. At the same time, we obtained the current density results of the microsystem, as shown in Figure 13, in which it can be observed that the current density was higher in the narrower power plane.

## 4. Discussion

In this paper, we systematically discuss our microsystem design and simulation methods. Simulation results show that shortening the length of the signal line and increasing the ground hole wraparound significantly increased the transmission quality of the signal lines. Widening the connecting lines of the power supply network reduced the DC voltage drop, and the division of the power supply plane affected its resonant mode. The temperature distribution was mainly caused by the chip power; therefore, temperature-sensitive devices should be kept away from high-power devices. Stress warpage results show that solder joints at the corners of the microsystem were subjected to the most severe deformation, and that signal lines should be avoided and used as redundant grounding.

## 5. Conclusions

This article describes how to design and simulate a system-in-package microsystem. We obtained adequate results during simulation analysis of signal integrity, power integrity, and thermal analysis. Critical signals, such as MCU signals, antennas’ RF signals, and signals between MCUs and IMUs, are required for microsystem operation. Analysis of the resonance characteristics and impedance characteristics of the power supply plane also showed satisfactory results. Thermal analysis showed that the reliability of the system is guaranteed.

## Figures and Tables

**Figure 1 micromachines-15-00167-f001:**
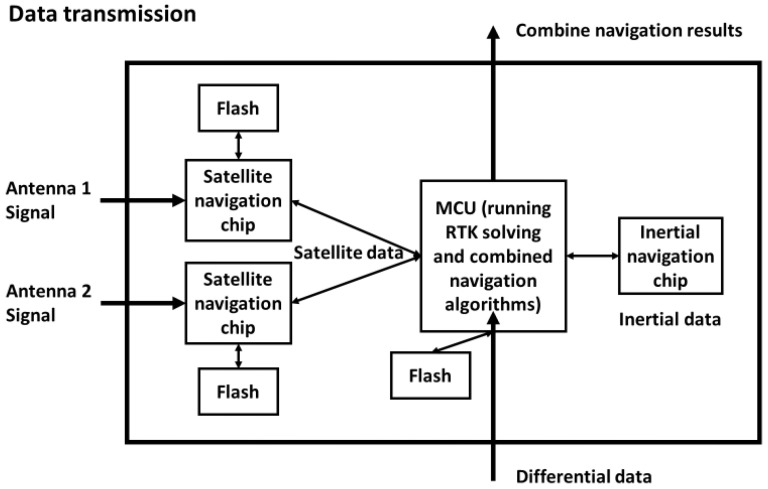
Schematic diagram of the combined navigation data transfer flow.

**Figure 2 micromachines-15-00167-f002:**
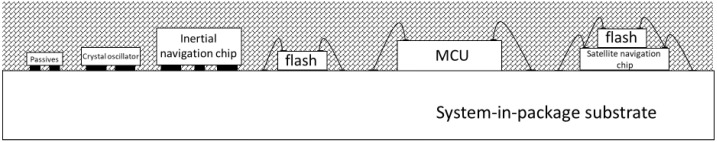
Schematic diagram of the microsystem packaging process.

**Figure 3 micromachines-15-00167-f003:**
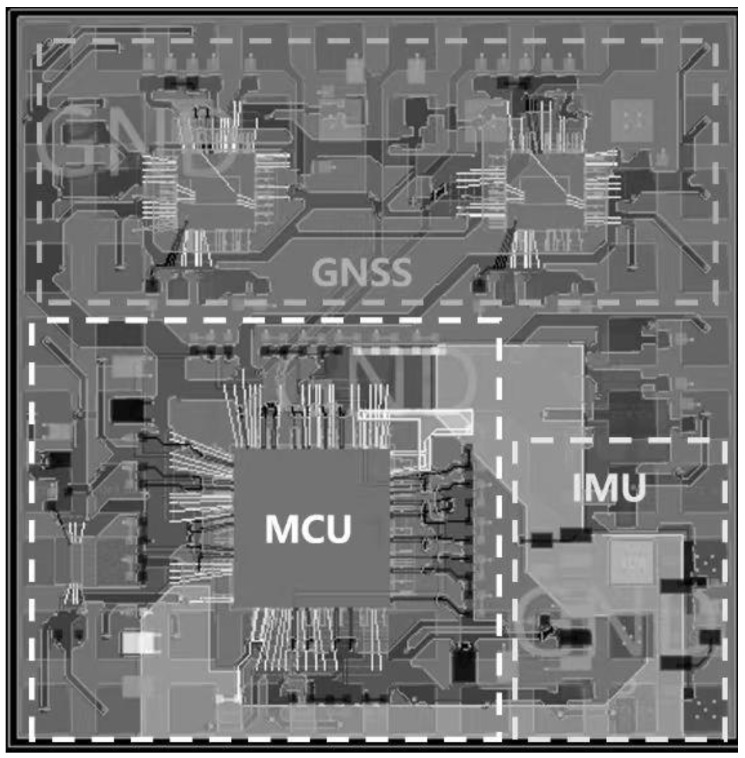
Layout of system-in-package chip.

**Figure 4 micromachines-15-00167-f004:**
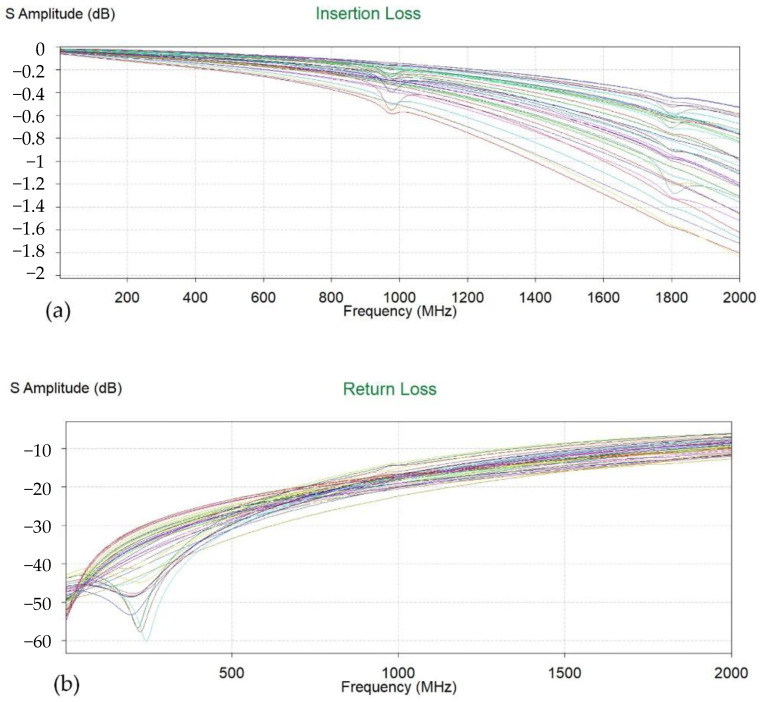
The S-parameter of MCU signals. (**a**) Insertion loss. (**b**) Return loss. The different colored lines represent different signal channels.

**Figure 5 micromachines-15-00167-f005:**
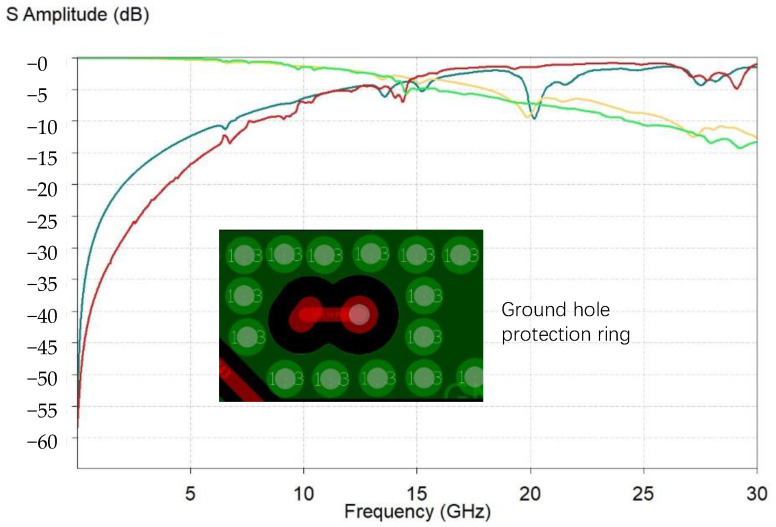
S-parameters of antenna signals. The red and blue lines represent the return loss, and the green and yellow lines represent the insertion loss.

**Figure 6 micromachines-15-00167-f006:**
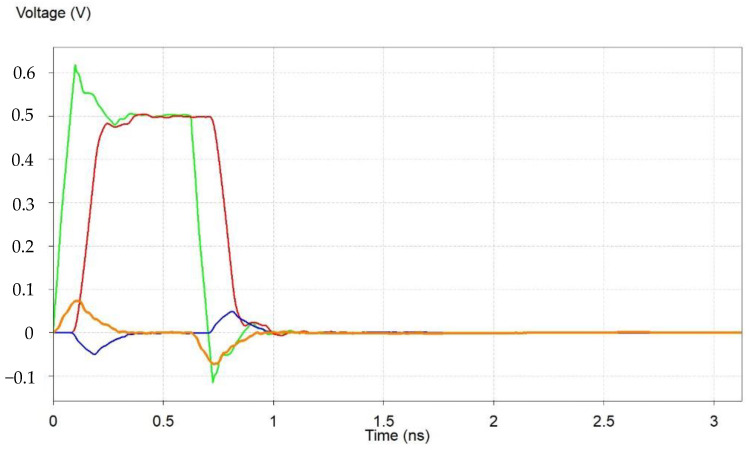
Time domain transient simulation results of the signals between MCU and IMU. The green line represents the TX signal, the red line represents the RX signal, the blue line represents the FEXT signal, and the orange line represents the NEXT signal.

**Figure 7 micromachines-15-00167-f007:**
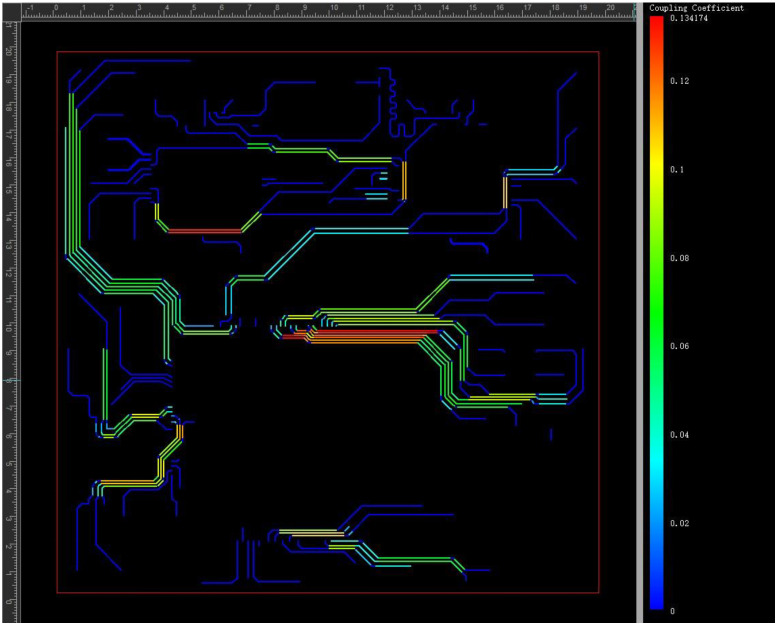
Coupling distribution of top traces.

**Figure 8 micromachines-15-00167-f008:**
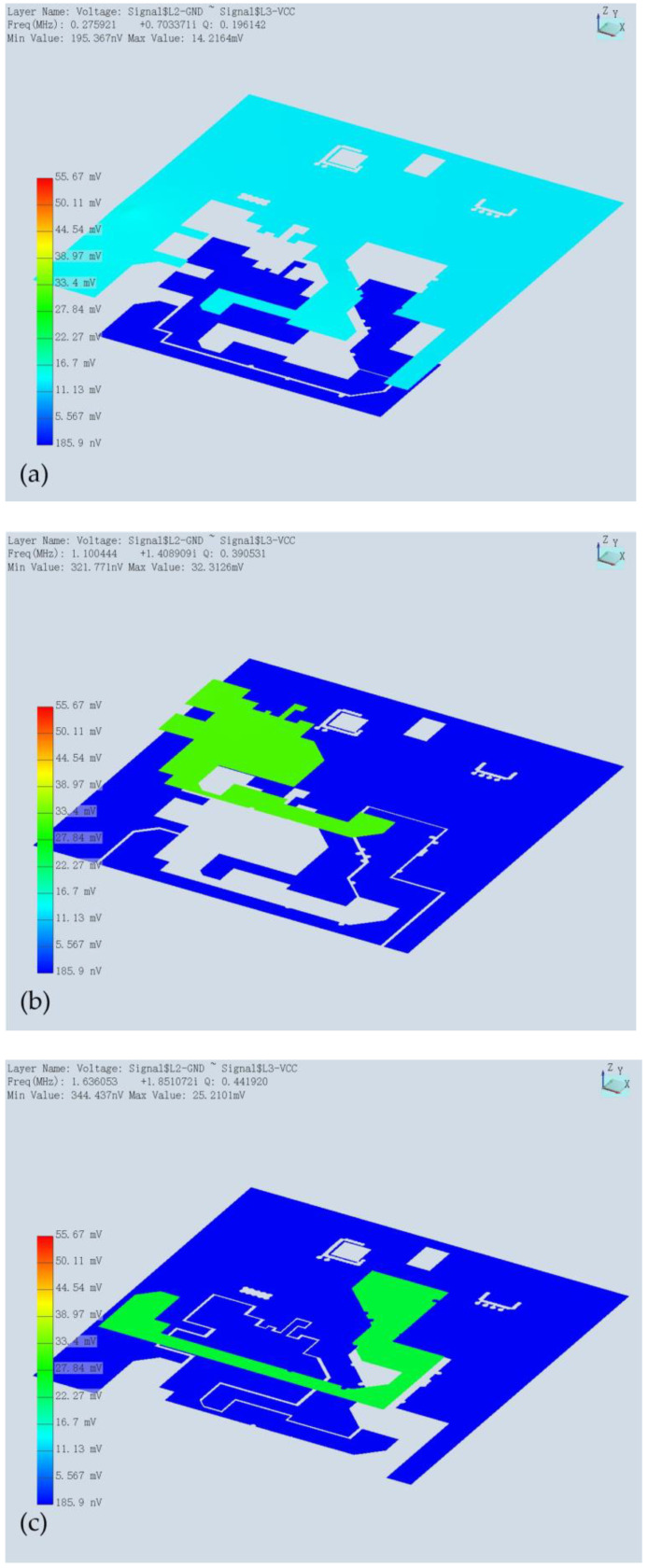
Resonance mode plots of the power plane. (**a**) First resonance. (**b**) Second resonance. (**c**) Third resonant. (**d**) Ninth resonance.

**Figure 9 micromachines-15-00167-f009:**
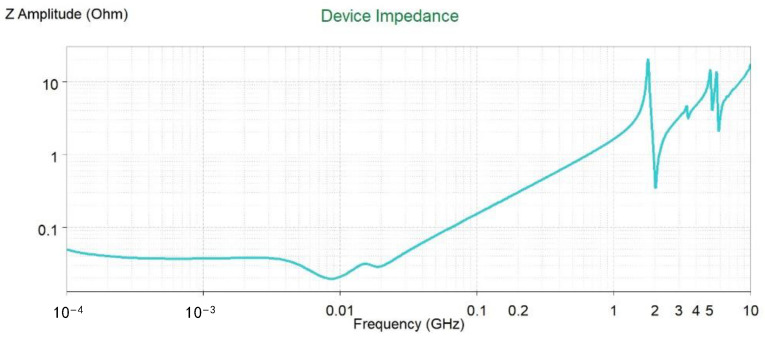
The 1.8 V impedance curve of the MCU.

**Figure 10 micromachines-15-00167-f010:**
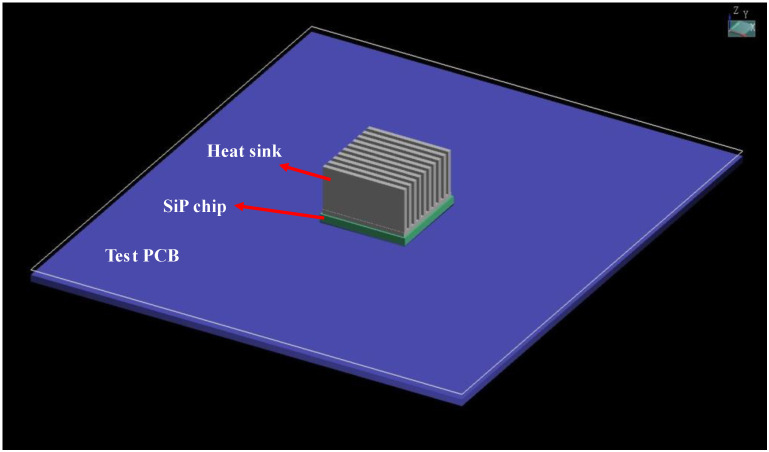
Thermal analysis scenario.

**Figure 11 micromachines-15-00167-f011:**
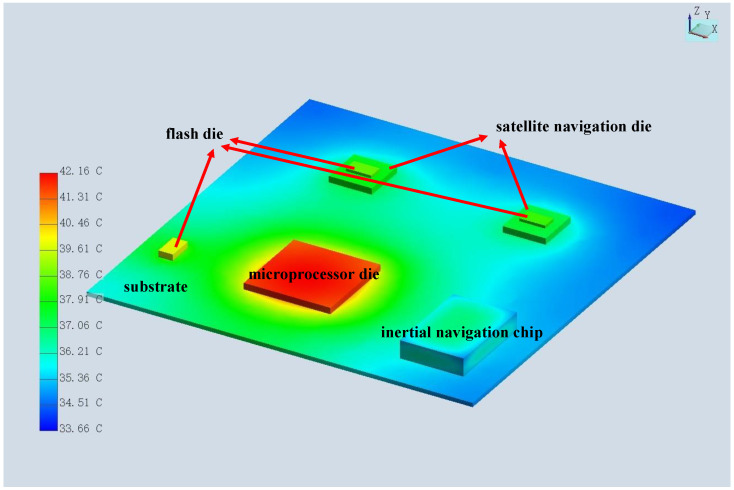
Temperature distribution of the system.

**Figure 12 micromachines-15-00167-f012:**
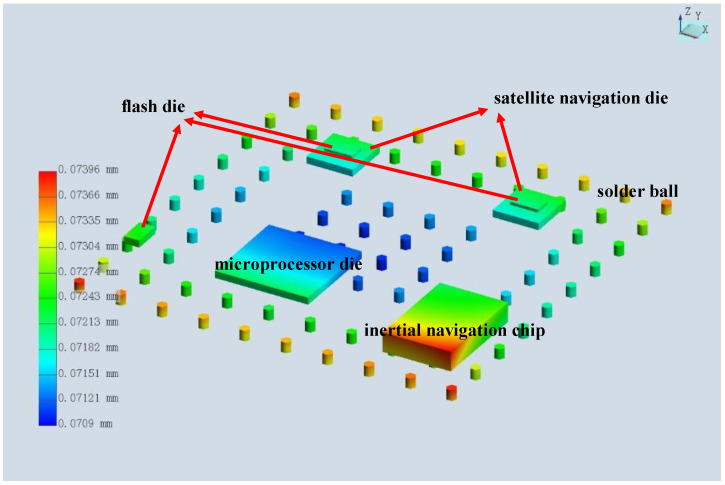
Warpage of the system due to thermal stress.

**Figure 13 micromachines-15-00167-f013:**
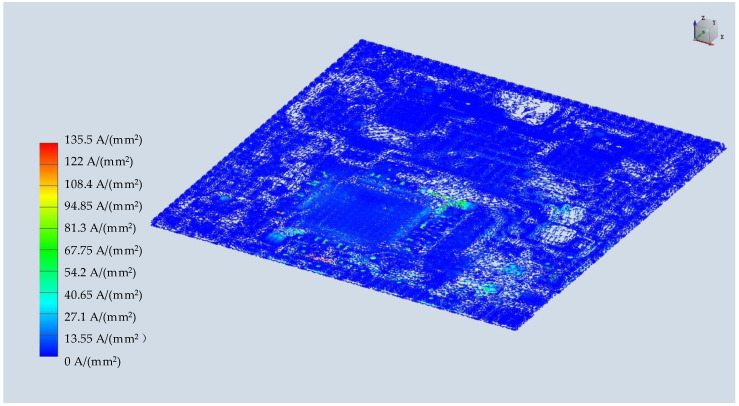
Current density distribution of the system.

**Table 1 micromachines-15-00167-t001:** Four-layer substrates’ stack information.

MYER NAME	THICKNESS (μm)		MYER NAME
Top solder mask	20	±10	AUS308 ^1^
M1 (Top)	15 (Min)	15 (Min)	Copper
PP	45	±15	GHPL-830NX(A) ^2^
M2	20	±5	Copper
CORE	60	±15	HL832NX(A) ^2^
M3	20	±5	Copper
PP	45	±15	GHPL-830NX(A)
M4 (Bottom)	15 (Min)	15 (Min)	Copper
Bottom solder mask	20	±10	AUS308
Finish	260	±40	

^1^ Taiyo Ink MFG, Hirasawa, Japan. ^2^ Mitsubishi Gas Chemical (MGC), Tokyo, Japan.

**Table 2 micromachines-15-00167-t002:** Resonant modes of the power plane.

No.	Resonance Freq (MHz)		Q Factor
1	0.275921	+0.703371i	0.196142
2	1.100444	+1.408909i	0.390531
3	1.636053	+1.851072i	0.44192
4	1.896135	+2.526094i	0.37531
5	1.911442	+2.527866i	0.378074
6	2.068185	+2.535850i	0.407789
7	2.076808	+2.443693i	0.424932
8	2.139904	+2.188735i	0.488845
9	3.528381	+3.044923i	0.579387

**Table 3 micromachines-15-00167-t003:** DC voltage drop characteristics of the power supply network.

Power Net	DC Voltage Drop	Percent (<2%)
1.8 V-MCU	17.6 mV	Pass
1 V-MCU	19.2 mV	Pass
3.3 V-MCU	17.5 mV	Pass
3.3 V-WD_A	7.1 mV	Pass
3.3 V-WD_B	6.1 mV	Pass
3.3 V-IMU	7.6 mV	Pass
3.3 V-FLASH	20.7 mV	Pass

**Table 4 micromachines-15-00167-t004:** Material and size parameters of key components.

Name	Main Material	Key Parameter
Heat sink	Copper	Height: 10 mm; Fin Width: 1 mm; Fin Pitch: 2 mm
SiP chip	Molding	Height: 1.76 mm; Width: 20 mm; Length: 20 mm
Test PCB	Copper, FR4	Height: 19.6 mm; Width: 114.5 mm; Length: 101.5 mm

## Data Availability

Data are contained within the article and Appendix A.

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
