# Peer review of "Design and Simulation of a System-in-Package Chip for Combined Navigation"

_micromachines, 2024, doi:10.3390/mi15020167_

Round 1

Reviewer 1 Report

Comments and Suggestions for Authors

The authors propose a review of the Design and simulation of System-in-package Chip for Combined navigation. The article is well organized and well written. There are a few minor comments that the authors should consider for better quality:

1.      Add results to the abstract showing that your design performs well in terms of signal, power and temperature, in terms of signal, power and temperature.

2.      Please add any abbreviations that are not listed anywhere.

3.      This article lacks a background introduction to such massive data. Writing background refers to the historical background in which the author created this article, and the context in which it was created. Therefore, please optimize the writing background of the article so that it can have a deeper understanding of the content.

4.      Add and clearly identify your contributions. In the introduction section of the article, the author's expression of the writing meaning is somewhat vague and difficult to understand. Please reorganize the language to express it so that readers can quickly understand the writing meaning and purpose of the article.

5.      In consideration of the paper structure, I propose a revision for Section 1 (Introduction) to enhance its clarity and conciseness.

6.       This article presents the research of several research methods. However, the literature section can highlight the shortcomings of various methods, which is enough to move on to proposed new methods. It is preferable to add the following references to enrich the work:

1.       https://doi.org/10.1016/j.aeue.2023.154980

2.       https://doi.org/10.3390/electronics12143154

3.       https://doi.org/10.4071/2015dpc-tp52

4.       https://doi.org/10.1109/cstic58779.2023.10219166

5        https://doi.org/10.1109/estc.2018.8546426

7.      Correct a few typing errors such as: SiP not SIP on line 24, add a dot to the title of Fig. 1 and Fig. 8, Add the unit of the value quoted on line 119.

8.      Please correct the title of Figure 11 so that it is on the same line and at the bottom of the Figure.

9.      Please add before Fig. 11 a figure showing the type of mesh used for these thermal results.

10.  Please identify the names of each chip listed in Fig. 12.

11.  Please add a table displaying the different parameters and layers simulated.

12.  Please refer to more recent references from the last five years.

Reviewer 2 Report

Comments and Suggestions for Authors

This manuscript proposes and designs a navigation microsystem device, and analyzes and studies the electrical, thermal, and signal integrity of the device. The manuscript has certain engineering reference value, but there is still a lot of room for improvement for research-oriented manuscript. It is recommended to conduct a major revision of the paper.

(1) The writing of scientific papers is not standardized enough, and this manuscript is more like a research report. It is recommended to use a more planned writing method to improve the overall quality of the manuscript.

For example, (1) the abstract lacks the results of the research work; (2) The figures in the manuscript need to be outputted with data before forming a graph, not a screenshot; (3) On the 51st line of the page 2 of the manuscript, "cross system is shown in the Figure below" should be changed to "cross system is shown in the Figure 1".

(2) The similarities and differences between the microsystem devices designed in this manuscript and those designed by previous researchers need to be increased, and the innovation of the devices designed in this paper should be indicated.

(3) The manuscript has conducted a single analysis of each physical field, and further discussion is needed to propose optimization plans or methods.

(4) Multiple physical fields often interact with each other. Should the author consider analyzing and exploring the coupling of multiple fields.

Comments on the Quality of English Language

The writing logic of the paper needs to be improved.

Round 2

Reviewer 2 Report

Comments and Suggestions for Authors

The author has made appropriate revisions based on the comments of the reviewers and agrees to accept and publish the manuscript.

Comments on the Quality of English Language

The quality of English writing needs to be improved.